

# Effects of high-risk human papillomavirus infection on P53, pRb, and survivin in lung adenocarcinoma—a retrospective study

Wenwen Sun[1,*], Hui Yang[1,*], Lu Cao[1], Ruochen Wu[1], Baoqi Ding[1], Xiaocui Liu[2], Xinli Wang[1] and Qiang Zhang[3]

[1] Department of Pathology, The Second Affiliated Hospital of Shandong First Medical University, Taian, Shandong, China
[2] Department of Histoloembryology, School of Clinical and Basic Medicine, Shandong First Medical University & Shandong Academy of Medical Sciences, Jinan, Shandong, China
[3] Shandong First Medical University, Jinan, Shandong, Taian
[*] These authors contributed equally to this work.

Corresponding authors
Xinli Wang, active1980@126.com
Qiang Zhang, chinazq007@163.com

## ABSTRACT

**Objective**. To observe the effects of high-risk human papillomavirus (HR-HPV) infection on P53, pRb, and survivin in lung adenocarcinoma (LUAD).

**Methods**. The cancerous and cancer-adjacent tissues of 102 patients with LUAD from January 2020 to April 2022 were selected for the study. HR-HPV infection was detected by flow fluorescence method, and P53, pRb, and survivin protein expression was detected by immunohistochemical staining method. Statistical analysis was performed to determine the differences in the HR-HPV infection and the expression of P53, pRb, and survivin proteins between LUAD tissues and cancer-adjacent tissues; the correlation between HR-HPV infection and P53, pRb, and survivin protein expression in cancer tissues; and the correlation between HR-HPV infection and clinicopathological features of LUAD.

**Results**. The infection rate of HR-HPV was higher in the LUAD tissues (28.43%) than in cancer-adjacent tissues (7.84%), and the difference was statistically significant ($P < 0.05$). The positive rates of P53 and survivin protein were higher in the LUAD group (33.33% and 67.16%, respectively) than in the cancer-adjacent group (3.92% and 11.73%, respectively), and the difference was statistically significant ($P < 0.05$). The positive rate of pRb protein was lower in the LUAD group (58.82%) than in the cancer-adjacent group (92.14%), and the difference was statistically significant ($P < 0.05$). The positive rates of P53 and survivin proteins were significantly higher in the HR-HPV LUAD group (58.62% and 86.21%, respectively) than in the non-HR-HPV LUAD group (41.38% and 67.12%, respectively), and the difference was statistically significant ($P < 0.05$). The expression rate of pRb protein was significantly lower in the HR-HPV LUAD group (37.93%) than in the non-HR-HPV LUAD group (67.12%), and the difference was statistically significant ($P < 0.05$). The expression of p53 and survivin protein was positively correlated with HR-HPV infection ($r = 0.338$ and 0.444, $P < 0.05$), whereas the expression of pRb protein was negatively correlated with HR-HPV infection ($r = -0.268$, $P < 0.05$). HR-HPV infection was not associated

with gender, age, and smoking in patients with LUAD ($P > 0.05$). HR-HPV infection was associated with lymph node metastasis and clinical stage of LUAD ($P < 0.05$). **Conclusions**. HR-HPV infection was associated with lymph node metastasis and clinical stage of LUAD, which may be achieved by up-regulating p53 and survivin protein expression and down-regulating pRb protein expression.

## INTRODUCTION

Human papilloma virus (HPV) is a known cause of human cancer, and has been shown to be associated with multiple sites of human cancer, such as cervix, reproductive tract, skin, head and neck, oropharynx and other sites (*Szymonowicz & Chen, 2020*; *De Martel et al., 2017*; *Serup-Hansen et al., 2014*; *Corredor et al., 2022*). The latest research shows that HPV infection plays a key role in the occurrence and development of Lung carcinoma (LA), especially in non small cell lung carcinoma (NSCLC) (*Zhou et al., 2022*). High risk human papillomavirus (HR-HPV) will produce HPV oncoproteins E6 and E7 after infecting cells. E6 protein can bind to p53, E7 protein can bind to retinoblastoma gene product (pRb), and lead to p53 mutation and pRb degradation. After DNA repair, angiogenesis and/or apoptosis and other complex processes, it finally leads to canceration (*Szymonowicz & Chen, 2020*). In HR-HPV infected squamous cell lesions, the up regulation of survivin expression is considered as an independent predictor, and there is also a correlation between HR-HPV infection and high survivin expression in Lung quadratus carcinoma (LSC) (*Wang et al., 2010*). Because of the high affinity between HPV and squamous cells, previous studies focused on the relationship between HPV and LSC. The relationship between HPV infection and lung adenocarcinoma (LUAD) was rarely reported. In order to explore whether HPV is related to the occurrence of LUAD and the possible mechanism, this paper analyzed the HPV-DNA level and p53, pRb, Survivin protein expression of LUAD patients by retrospective clinical study of LUAD patients. To explore the correlation between p53, pRb, Survivin protein expression and HR-HPV infection in LUAD patients.

## METHODS

### Subjects

The paraffin specimens of cancerous and cancer-adjacent tissues of 102 patients with invasive LUAD admitted at the Second Affiliated Hospital of Shandong First Medical University from January 2021 to April 2022 were collected for the study. The study was approved by the ethics committee of The Second Affiliated Hospital of Shandong First Medical University (ID: 2021-8). Inclusion and exclusion criteria: (1) Inclusion criteria: The diagnosis of LUAD was in accordance with WHO pathological diagnosis norms; patients were untreated and had complete clinical data. (2) Exclusion criteria: Patients with other malignant tumors and past history of invasive LUAD were excluded. All specimens were used with written consent from participants or family members.

## Data collection

Basic information of patients: 40 males and 62 females; age: 25 patients < 60 years old, 77 patients ≥60 years old, with an average age of 63.02 ± 8.08 years; smokers: 28 patients, non-smokers: 74 patients; lymph node metastasis: 11 patients, no metastasis: 91 patients; tumor diameter < 3 cm: 87 patients, ≥3 cm: 15 patients; clinical stage I+II: 93 patients, III+IV: nine patients.

## Detection of HPV infection by flow fluorescence-liquid chip technology

Two 7 μm slices of tissue wax blocks were used to extract DNA, and 5 μl of DNA template was added to each 15 μl PCR reaction system to carry out PCR amplification reaction; 3 μl of amplification product and 22 μl of microsphere hybridization solution were added to each well of the microwell hybridization plate for denaturation hybridization (95 °C, 5 min; 48 °C, 30 min).Then, 75 μl of phycoerythrin (SA-PE) was added to each well to continue hybridization (48 °C, 15 min), and the HPV infection data were read on the Luminex 200 platform. A Globin value ≥150 was considered as a successful test, and positive high-risk types were counted in the results (types 16, 18, 31, 33, 35, 39, 45). The kit was purchased from Shanghai Toujing Life Technology Co., LTD.

## Detection of P53, pRb, and survivin expression in tissues by immunohistochemical staining method

Tissue paraffin blocks were sliced at 3 μm, incubated at 75 °C for 60 min, routinely deparaffinized to water, and rinsed for threetimes with phosphate buffer solution (PBS). Antigen retrieval was performed by using pH9.0 EDTA solution at high temperature and high pressure for 2 min. The samples were rinsed with PBS for threetimes, and 3% $H_2O_2$ was added to eliminate endogenous peroxidase. Then, the samples were rinsed for threetimes with PBS. Primary antibodies were added (P53 mouse monoclonal antibody, pRb mouse monoclonal antibody, survivin rabbit polyclonal antibody; all three antibodies were ready-to-use; Fuzhou Maixin Biotechnology Development Co., Ltd., Fuzhou, China). The samples were incubated for 90min at room temperature and then rinsed for threetimes with PBS. Secondary antibody (ready-to-use Quick MaxVisionTM-HRP kit (mouse/rabbit)) was added dropwise and incubated for 15 min at room temperature. Then, it was rinsed with PBS for threetimes and stained with DAB for 3–5 min. Hematoxylin was used for staining cell nuclei, and then conventional dehydration, transparency, and sealing of the film.

## Interpretation criteria of immunohistochemical staining results

The results were interpreted by two senior pathologists in a double-blind method. Judgment criteria: ① P53-positive location was in the nucleus, which was diffusely positive (positive rate ≥70%) or completely negative (positive rate 10%). ② pRb-positive localization was in the nucleus. pRb staining intensity (0 = unstained, 1 = weakly stained, 2 = moderate staining, 3 = strong staining) and the range of positive expression were recorded at the same time (score: 0, <5%; 1,5%–25%; 2, 26%–50%; 3, 51%–75%; 4, 76%–100%). The staining intensity score was multiplied by the positive expression range of pRb to obtain

a composite expression score (CES). According to CES, it was divided into RB1 negative (or low expression, CES<6) and positive (high expression, CES $\geq$6). ③ Survivin-positive location was in the cytoplasm/nucleus, and the positive localization area was stained with brown-yellow particles under the microscope. Fivefields of high magnification ($\times$200) were randomly selected. According to the coloring intensity score, 0 was no coloring, 1 was light yellow, 2 was yellow, and 3 was brown yellow. Meanwhile, according to the percentage of positive cells in the same type of cells, 0 was negative, 1 indicated < 10% positive cells, 2 indicated 10%–50% positive cells, 3 indicated 50%–75% positive cells, and 4 indicated >75% positive cells. The sum of the two scores was taken as the total score, in which 0–3 was negative and >3 was positive.

### Experimental grouping

The samples were divided into the LUAD group and cancer-adjacent group. The LUAD group was further divided into HR-HPV LUAD group and non HR-HPV LUAD group according to the presence or absence of HR-HPV infection. Our study obtained the consent of the patients and their families, and was approved by the hospital ethics committee, the ethics approval number is 2021-8.

### Statistical analysis

SPSS22.0 software was used to analyze and process the data (SPSS Inc. Chicago, IL, USA). The count data were expressed as the number of cases or percentage, and the $X^2$ test or Fisher's exact probability method was used between groups. Measurement data were expressed as ($\bar{X}\pm$s) between groups. The Spearman correlation coefficient was used for correlation analysis. $P < 0.05$ was considered to indicate statistically significant differences.

## RESULTS

### Comparison of HR-HPV infection rate between LUAD group and cancer-adjacent group

The HR-HPV infection rate was higher in the LUAD group than in the cancer-adjacent group, and the difference was statistically significant ($P < 0.05$) (Table 1).

### Correlation analysis between HR-HPV infection and clinicopathological characteristics of LUAD

HR-HPV infection in LUAD was related to lymph node metastasis and clinical stage. With the improvement of lymph node metastasis and clinical stage, the detection rate of HR-HPV was increased, and the difference was statistically significant ($P < 0.05$). HR-HPV infection was not associated with gender, age, and smoking, and the difference was not statistically significant ($P > 0.05$) (Table 2).

### Expression of P53, pRb, and survivin in the LUAD group and cancer-adjacent group

The positive rate of P53 and survivin was higher in cancer tissues than in cancer-adjacent tissues, and the difference was statistically significant ($P < 0.05$). The positive rate of pRb was lower in the LUAD group than in the cancer-adjacent group, and the difference was statistically significant ($P < 0.05$) (Table 3).
**Table 1  Comparison of HR-HPV infection rate between LUAD group and paracancerous group ($n = 102$).**

| Group | HR-HPV infection | | $X^2$ | $P$-value |
|---|---|---|---|---|
| | + | − | | |
| LUAD group | 29 | 73 | 14.56 | $P < 0.05$ |
| cancer-adjacent group | 8 | 94 | | |

**Table 2  Correlation analysis between HR-HPV infection and clinicopathological characteristics of LUAD ($n = 102$).**

| Features | | Number | HR-HPV infection | | $X^2$ | $P$-value |
|---|---|---|---|---|---|---|
| | | | Positive | Negative | | |
| Gender | Male | 40 | 10 | 29 | 0.617 | $P > 0.05$ |
| | Female | 62 | 19 | 42 | | |
| Age | <60 years | 31 | 7 | 24 | 0.749 | $P > 0.05$ |
| | ≥60 years | 71 | 22 | 49 | | |
| Smoking status | Yes | 28 | 10 | 18 | 1.006 | $P > 0.05$ |
| | No | 74 | 19 | 55 | | |
| Lymph node metastasis | Yes | 11 | 8 | 3 | 11.89 | $P < 0.05$ |
| | No | 91 | 21 | 70 | | |
| Stage | I+II | 93 | 27 | 66 | 8.774 | $P < 0.05$ |

**Table 3  Expression of P53, pRb, and survivin in the LUAD group and cancer-adjacent group ($n = 102$).**

| Correlation factor | LUAD group | | Cancer-adjacent group | | $X^2$ | $P$-value |
|---|---|---|---|---|---|---|
| | Number | Ratio of positive (%) | Number | Ratio of positive (%) | | |
| P53 | 34 | 33.33 | 4 | 3.92 | 22.25 | $P < 0.05$ |
| pRb | 60 | 58.82 | 95 | 92.14 | 32.90 | $P < 0.05$ |
| survivin | 63 | 61.76 | 12 | 11.76 | 52.43 | $P < 0.05$ |

## Correlation between HR-HPV infection and expression of P53, pRb, and survivin in LUAD

The positive rates of P53 and survivin were higher in the HR-HPV adenocarcinoma group than in the non HR-HPV LUAD group, and the difference was statistically significant ($P < 0.05$). The positive rate of pRb was lower in the HR-HPV LUAD group than in the non-HR-HPV LUAD group, and the difference was statistically significant ($P < 0.05$). The expression of p53 and survivin was positively correlated with HR-HPV infection, whereas the expression of pRb was negatively correlated with HR-HPV infection (Fig. 1, Tables 4 and 5).

## DISCUSSION

HPV is an uncoated, double-stranded, epithelial DNA virus (*Psyrri, Boutati & Karageorgopoulou, 2011*), with more than 150 types, which can be transmitted through skin and/or sexual contact. There are three hypotheses for the pathogenesis of intrapulmonary
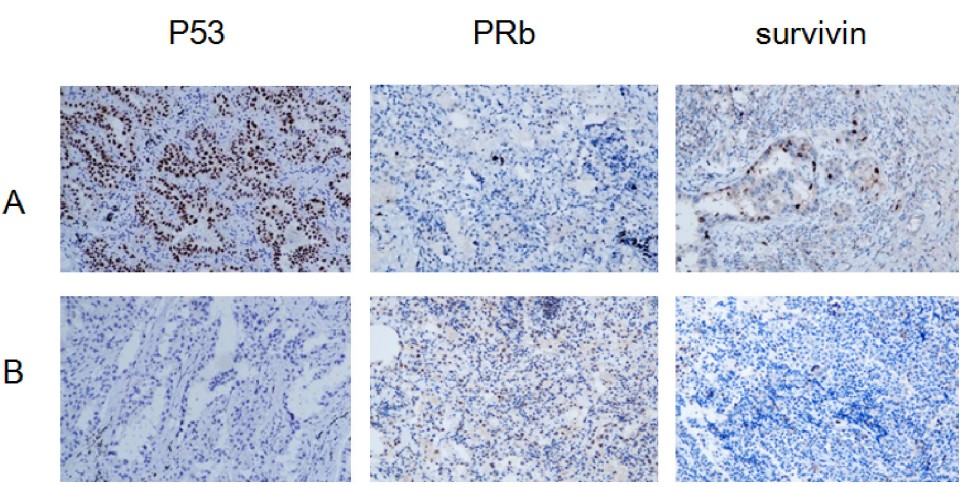

**Figure 1** Expression of P53, PRb and surviving: HR-HPV LUAD group (A), non- HR-HPV LUAD group (B), IHC ×200.

**Table 4  Correlation between HR-HPV infection and expression of P53, pRb, and survivin in LUAD.**

| correlation factor | HR-HPV LUAD group ($n = 29$) | | Non- HR-HPV LUAD group ($n = 73$) | | $X^2$ | $P$-value |
|---|---|---|---|---|---|---|
| | Number | Ratio of positive (%) | Number | Ratio of positive (%) | | |
| P53 | 17 | 58.62 | 4 | 41.38 | 35.82 | $P < 0.05$ |
| pRb | 11 | 37.93 | 49 | 67.12 | 7.302 | $P < 0.05$ |
| survivin | 25 | 86.21 | 38 | 52.05 | 10.25 | $P < 0.05$ |

**Table 5  Relationship between P53, pRb, and survivin expression and HR-HPV infection in LUAD group.**

| Positive rate | HR-HPV infection | |
|---|---|---|
| | $r$ | $P$-value |
| P53 | 0.338 | $P < 0.05$ |
| pRb | −0.268 | $P < 0.05$ |
| survivin | 0.444 | $P < 0.05$ |

HPV infection: (1) through cervical lesions to the lungs; (2) through mouth-to-genital high-risk sexual transmission, where HPV travelled from the infected genital system to the mouth,throat,then to the lungs; (3) through the air and respiratory system to the lungs (*Xiong et al., 2017*). Most HPV types with potential carcinogenicity are high-risk, and HPV16 and HPV18 have the highest detection rate (*Magalhães et al., 2021*). The positive rate of HPV infection in LA varies greatly in different studies. The difference of positive rate may be related to histopathological type, detection method, sample source and other factors. According to statistics, the positive rate of HPV in LA tissues in Asia is 28.1–35.7% (*Klein, Amin Kotb & Petersen, 2009*; *Hasegawa et al., 2014*). In this study, 29 of 102 LUAD patients were positive for high-risk HPV infection, with a positive rate of 28.43%, which is

consistent with previous studies, indicating that HR-HPV infection does exist in LUAD and accounts for a certain proportion. There is still controversy about whether HPV infection is related to smoking. Statistics show that non-smokers infected with HPV have a higher risk of LA than smokers (*Cheng et al., 2008*). However, this study showed that high risk HPV infection LUAD was not related to smoking, and the differences in research results might be related to different sensitivity and specificity of detection methods, population differences and sample size.

HPV oncogenes E6 and E7 are always expressed in HPV positive tumors, which are necessary to maintain the transformation phenotype of HPV positive tumor cells. In HR-HPV induced epithelial tumors, the E6 oncoprotein is a key determinant of cell transformation, which induces the degradation of apoptosis promoting tumor suppressor p53 in the host. E6 activates the intracellular ubiquitin ligase E6AP, which then induces proteasome dependent p53 degradation. In this process, E6 binds to the consensus sequence of short LxxLL rich in leucine (L) in the cell ubiquitin ligase E6AP, and then E6/E6AP heterodimer degrades p53 lazily. Degradation of p53 is a marker activity of HPV E6 protein (*Martinez-Zapien et al., 2016*; *Li et al., 2019*). P53 gene is an important tumor suppressor gene, which plays a role in inhibiting tumor by coordinating DNA repair, cell cycle arrest, cell aging, death, differentiation, metabolism and other cellular reactions (*May & May, 1999*). A lot of research has been done on the relationship between p53 gene and tumor, and many tumors are related to p53 gene mutation. Nucleophosphoprotein P53 is composed of 393 amino acids, with a short half-life, which is difficult to detect by immunohistochemistry (*Zhang et al., 2020*). When p53 mutation occurs, aggregation occurs and function is lost, leading to negative growth of protein function, promoting abnormal cell proliferation, tumor progression and anti apoptosis. Due to lack of normal physiological characteristics, it is more stable and has a longer half-life, which can be detected by immunohistochemistry (*Yu et al., 2018*; *Kanapathipillai, 2018*). The results of this study showed that the positive rate of P53 in LUAD was 33.33%, which was significantly higher than the positive rate of adjacent tissues 3.92%, indicating that p53 gene mutation played a role in LUAD. The positive rate of LUAD P53 in HR-HPV infection was 58.62%, which was significantly higher than that in HR-HPV negative LUAD P53 infection (41.38%). After correlation analysis, it was found that the positive rate of p53 protein was significantly and positively correlated with HR-HPV infection. It was speculated that in HR-HPV infection of LUAD, normal wild type p53 was degraded by HPV cancer protein E6, and the development of LA was promoted synergistically with P53 mutation.

PRb is the protein of tumor suppressor that was first discovered and determined by human beings. Its family members, pRb/p105, p107 and pRb2/p130, are one of the most important regulators of G1/S transition in the cell cycle. The point mutation or inactivation deletion of Rb gene leads to the loss of protein regulation ability, which is related to the occurrence and development of many tumors (*Mandigo et al., 2021*). At the same time, Rb family is an important target for virus replication and virus induced mammalian cell transformation. By interfering with the growth mechanism of infected cells, viruses force them to replicate DNA to complete their own replication. The interaction between early viral antigens and cell cycle regulators is an important mechanism for viruses to remove

cell cycle regulation and lead to cell transformation. Early virus T antigen can bind all members of the pRb family, promote the activation of the transcription factor E2F family, and thus induce the expression of genes required for entering the S phase (*Caracciolo et al., 2006*). HPV E7 targets to destroy pRB and lose protein stability by binding pRB, leading to the degradation and functional inactivation of pRb, and blocking the cell cycle in G0/G1 phase (*Hoppe-Seyler et al., 2018*). The results of this study showed that the positive rate of LUAD pRb was 58.82%, which was lower than 92.14% in the adjacent group, indicating that pRb played a certain role in the occurrence of LUAD. The positive rate of LUAD pRb in HR HPV infection was 37.93%, which was significantly lower than that in HR HPV negative LUAD pRb infection (67.12%). After correlation analysis, it was found that the positive rate of pRb3 protein was significantly negatively correlated with high-risk HPV infection. It was speculated that HPV oncoprotein E7 could target to destroy pRb and reduce its expression in LUAD infected by HR HPV, and promote the development of LA together with pRb mutation.

Survivin, a member of the inhibitor of apoptosis protein family, is composed of 142 amino acids. Its multiple residues can bind with caspase, directly inhibit its activity, accelerate cell proliferation, and resist apoptosis. Survivin directly regulates apoptosis and mitosis of tumor cells during tumor occurrence and metastasis. Survivin is highly expressed in cancer tissues of patients with malignant tumors, and is the strongest inhibitor of apoptosis protein found so far (*Beltran et al., 2016*; *Nakamura et al., 2018*). Survivin may participate in HPV mediated differentiation of normal squamous epithelium by regulating the process of cell apoptosis. The expression of survivin is related to the reduction of overall survival rate of malignant tumors (*Lo Muzio et al., 2004*). In an experiment to evaluate the viability and cell cycle of HPV16 positive cancer cells after knockout of survivin by small interfering RNA (si survivin), it was found that targeted survivin expression may be one of the treatment methods for cancer caused by HR-HPV infection (*Lin et al., 2020*). This study showed that 61.76% of LUAD tissues had survivin expression, which was significantly higher than that of adjacent tissues. It shows that survivin gene is up-regulated in LUAD and participates in the occurrence and development of LUAD by inhibiting the apoptosis of LUAD cells. In the correlation analysis, we found that the expression of survivin in LUAD was significantly correlated with the expression of HR-HPV infection. It indicates that there is a direct or indirect relationship between HR-HPV infection and survivin expression in LUAD, and survivin may become a therapeutic target for treating HR-HPV infected LUAD.

In the correlation analysis between the clinicopathological characteristics of LUAD patients and HR-HPV, it was found that HR-HPV infection was related to the clinical stage of LUAD and whether lymph nodes were metastatic. The proportion of HR-HPV infected LUAD with high clinical stage and the proportion of lymph node metastasis were higher than those of HR-HPV negative LUAD, suggesting that HR-HPV infection is a high risk factor for LUAD, and its mechanism may be involved in the occurrence and development of LUAD by regulating the expression of P53, pRb, and survivin. P53 mutation plays an important role in the therapeutic drug resistance of cancer, and p53 related signal pathway may reverse the process of cisplatin resistance (*Zhang et al., 2020*). PRb deficient tumors

have more favorable response to specific chemotherapy. The regulatory effect of HR-HPV on P53 and pRb in the treatment of LUAD deserves attention.

It is worth mentioning that the number of cases in the middle and late stages in this study is relatively small, which may be related to the widespread application of screening methods such as electronic computer tomography and the early detection of LA (*Schabath & Cote, 2019*). This is a retrospective study. Due to the limitation of the number of cases, this study has some limitations, and the results may be biased. In the next step, we can increase the sample size for further study.

### Funding
This work was supported by the Tai'an Science and Technology Innovation Development Project (No. 2021NS231). The funders had no role in study design, data collection and analysis, decision to publish, or preparation of the manuscript.

### Grant Disclosures
The following grant information was disclosed by the authors:
Tai'an Science and Technology Innovation Development Project: No. 2021NS231.

### Competing Interests
The authors declare that there are no competing interests.

### Author Contributions
- Wenwen Sun performed the experiments, prepared figures and/or tables, authored or reviewed drafts of the article, and approved the final draft.
- Hui Yang performed the experiments, prepared figures and/or tables, authored or reviewed drafts of the article, and approved the final draft.
- Lu Cao performed the experiments, prepared figures and/or tables, and approved the final draft.
- Ruochen Wu performed the experiments, prepared figures and/or tables, and approved the final draft.
- Baoqi Ding performed the experiments, prepared figures and/or tables, and approved the final draft.
- Xiaocui Liu performed the experiments, prepared figures and/or tables, and approved the final draft.
- Xinli Wang conceived and designed the experiments, analyzed the data, authored or reviewed drafts of the article, and approved the final draft.
- Qiang Zhang conceived and designed the experiments, analyzed the data, prepared figures and/or tables, authored or reviewed drafts of the article, and approved the final draft.

### Human Ethics
The following information was supplied relating to ethical approvals (i.e., approving body and any reference numbers):

The ethics committee of Second Affiliated Hospital of Shandong First Medical University.

## Ethics

The following information was supplied relating to ethical approvals (i.e., approving body and any reference numbers):

Ethics committee of The Second Affiliated Hospital of Shandong First Medical University.

## Data Deposition

The raw measurements are available in the Supplemental File.

## Supplemental Information

Supplemental information for this article can be found online at http://dx.doi.org/10.7717/peerj.15570#supplemental-information.

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
