# Peer review of "Effects of high-risk human papillomavirus infection on P53, pRb, and survivin in lung adenocarcinoma—a retrospective study"

_PeerJ, doi:10.7717/peerj.15570_

## Round 0.1 · original submission · Minor Revisions

Please carefully read the comments and suggestions from the reviewers and provide your point-by-point responses.

Reviewer 1 ·

Basic reporting

In this study, Sun et al. reported the molecular effects of papillomavirus(HR-HPV) infection on the expression of P53, pRb, and surviving in the setting of lung adenocarcinoma. From the comprehensive analysis of large cohort of human patients and control samples, authors found that P53 and survivin proteins were significantly higher in the HPV LUAD lungs. In contrast, The expression rate of pRb protein was significantly lower in the HPV LUAD. As a result, P53/Survivin and pRb are positively and negatively correlated with HR-HPV infection, respectively. This study provides novel insights into the pathogenesis underlying LUAD with HR-HPV infection, revealing an important risk factor of viral infection of LUAD that are previously overlooked in the field. The experiments are well designed and performed to high standard. The data are solid, and presented in a logic manner. The conclusion is well supported by the current analysis. I have no further concerns regarding the manuscript, and suggest the next step toward publication.

Experimental design

no comment

Validity of the findings

no comment

Additional comments

no comment

Reviewer 2 ·

Basic reporting

1.I cannot find table 5. Authors need to carefully check if they missed uploading table 5 in the submission portal.
2.Fig 1: My question is which group does A represent, and which group does B represent? A bar plot to quantify the staining intensity needs to be plotted in order to show the readers a clear understanding about the protein expression level of three genes between experimental samples and control samples.
3.Introduction: How about the study progress of this topic? Is there anybody else having done such related study? Is there any research gap that authors wish to address? How about the innovation point of your research? The introduction section is too simple.
4.The subheading “2.5Interpretation of results” needs to be changed. This subheading title is not appropriate.
5.The study design of the retrospective study needs to be described in detail.
6.My concern is if there are some findings in your research contradictory with the others’ finding in the previous literature? If so, please discuss it and explain the reasons.
7.This manuscript needs to be carefully proofread by a fluent English speaker in the medical field.

Experimental design

no comment

Validity of the findings

no comment

Additional comments

no comment

Reviewer 3 ·

Basic reporting

The relationship between HR-HPV and LUAD is a hot topic in clinical research, and the authors of this manuscript found that P53, pRb, and survivin may be involved in the pathogenesis. The authors collected clinicopathological data and made a detailed study, which is helpful to the clinical understanding of LUAD.

Experimental design

In this manuscript, the authors observed only P53, pRb, and survivin expression changes in tumor tissues and adjacent tissues, and analyzed them based on lymph node metastasis and clinical tumor staging. This study lacks in-depth mechanism research, but as a clinical study, it is basically OK.

Validity of the findings

The data in this manuscript are detailed and reliable. The authors conclude that HR-HPV infection is associated with lymph node metastasis and clinical staging of LUAD, possibly by up-regulating p53 and survivin protein expression and down-regulating pRb protein expression. This conclusion was only found in patients diagnosed with clinical tumors, and its role in disease progression and prevention remains to be further explored.

Additional comments

1. P10, the authors mentioned Table 4 and 5, but there is only table 4 in the paper. Please provide Table 5.
2. Please provide scale for Figure 1
3. Smoking status did not affect the positive rate of HR-HPV, but it did affect LUAD. Please refer to previous literature reports in the discussion to explain: Do smoking and HR-HPV have a synergistic effect on the incidence of LUAD?

---

## Round 0.2 · accepted · Accept

The authors have addressed all the comments from reviewers.